# The Impact of Physical Activity on Long COVID Symptoms Among College Students: A Retrospective Study

**DOI:** 10.3390/ijerph22050754

**Published:** 2025-05-11

**Authors:** Gili Joseph

**Affiliations:** Science Faculty, Kibbutzim College of Education, 149 Namir Drive, Tel-Aviv 6250769, Israel; gilijosephphd@gmail.com or gili.joseph@smkb.ac.il

**Keywords:** low levels of physical activity, high levels of physical activity, long COVID, COVID-19, undergraduate students

## Abstract

Millions worldwide suffer from long COVID, which affects daily life and impairs multiple organs. Younger adults report symptoms more frequently than older adults. Since physical activity enhances overall health, this study examines whether regular exercise reduces long COVID severity in college students. This cross-sectional retrospective study surveyed 309 teacher-training college students about their long COVID symptoms and physical activity levels. Participants were categorized based on activity levels, and symptom differences were analyzed. Among respondents, 44 (14.4%) reported long COVID symptoms, with fatigue being the most common (13.3%). Students engaging in regular, intense physical activity did not experience fewer symptoms than less active students (1.83 ± 0.85; 1.75 ± 0.89, *p* = 0.376). However, physical education students reported fewer symptoms than students in other programs (6.7% vs. 17.4%). Greater self-reported participation in physical activity was not associated with less reported long COVID symptoms among college-aged students; however, students enrolled in physical education programs—who integrate physical activity into their daily routines as part of their academic curriculum—reported fewer symptoms, suggesting that sustained, structured physical activity may contribute to reduced symptom burden. Further research is needed to explore this relationship.

## 1. Introduction

As of August 2024, the COVID-19 pandemic has resulted in the deaths of more than 7 million people worldwide, with nearly 800 million individuals having contracted the disease, experiencing symptoms ranging from asymptomatic to extremely severe [1]. Although the pandemic was defined as having ended, the estimation is that 10–35% of the convalescents who were not hospitalized and 50–70% of the hospitalized COVID-19 patients suffer from long-term symptoms [2,3]. An individual is classified as experiencing long COVID if symptoms persist for more than three months following the initial COVID-19 infection and last for at least two months [4].

There are many symptoms of long COVID which manifest in many ways in various organs and areas of the body. The most common symptoms are fatigue, dyspnea, concentration problems, decreased exercise capacity, and sleep disorders [2,3,5,6]. Interestingly, adults under 50 report a higher prevalence of long COVID symptoms compared to those over 50, and evidence suggests that greater severity of symptoms during the acute phase of COVID-19 is associated with an increased likelihood of developing long COVID [2].

Physical activity improves multiple physiological systems. It improves metabolic functions, enhances endurance, and supports healthy weight maintenance [7]. It also reduces the risk of depression and anxiety [8,9], and may strengthen immune responses, potentially protecting against severe infections and prolonged recovery such as in long COVID [10,11,12]. The various health benefits of regular physical activity suggest that they might help manage long COVID symptoms like autonomic dysfunction and fatigue [13,14], as well as reduce anxiety and depression while improving cognitive function, potentially supporting long COVID patients experiencing neurocognitive symptoms [15,16]. Moreover, studies suggest that physical activity during the pandemic reduced the likelihood and duration of long COVID symptoms [17,18], and pre-infection physical activity was linked to a lower risk of post-COVID self-care difficulties [19]. In addition, individuals engaging in higher physical activity levels before COVID-19 had a lower risk of severe COVID-19 outcomes [20]. While regular physical activity may offer benefits for both acute and long-term COVID-19 symptoms in non-hospitalized individuals, current findings remain inconclusive due to the complex nature of long COVID and heterogeneity in study methodologies [16,17,18,19,20,21]. Notably, young adults have been found to report long COVID symptoms more frequently than older populations (2). However, the college student population—representing a significant subset of young adults—has been relatively underexplored in the context of long COVID, despite their vulnerability and the potential impact on academic functioning and quality of life. This study therefore aims to examine the prevalence and characteristics of long COVID among college students. Additionally, it investigates whether differences exist between students enrolled in general academic programs and those studying physical education, a discipline in which physical activity is an integral part of the curriculum, resulting in higher frequency and intensity of exercise among the latter group.

## 2. Materials and Methods

### 2.1. Study Design and Population

This cross-sectional study was conducted between December 2022 and March 2023. During this period, a survey was distributed via Google Forms to 2830 students enrolled in a Bachelor of Education (B.Ed.) program at a college of education. This study aimed to investigate the prevalence, symptom characteristics, and impact of physical activity on existing long COVID among college-aged students. The study also aimed to explore whether differences in long COVID symptom prevalence exist between students in general academic programs and those in physical education programs, where physical activity is systematically integrated into the curriculum and performed regularly. Inclusion criteria: Participants were included if they were enrolled at the college and were 18 years or older. Individuals were excluded if they were not enrolled at the college, did not complete the questionnaire, or were under 18 years old.

The study was approved by the ethics committee of the institute (approval # SHC2022_16), and a written consent was obtained from all study participants.

### 2.2. Questionnaire and Variables:

For the physical activity questions, and for the long COVID symptoms and acute illness phase symptoms, we used validated questionnaires from published studies [22,23]. All questionnaires were self-reported and distributed via digital delivery.

### 2.3. Physical Activity

Physical activity was assessed using three specific parameters. Each participant was required to respond to three questions related to their physical activity:

(1). Type of Physical Activity: Participants were asked to indicate the type (s) of physical activity they engaged in, with options provided including indoor or outdoor activities, walking, aerobic exercises (such as running, cycling, rowing, etc.), strength training, yoga/Pilates, dancing, tennis, ball games, and other sport classes or activities.

(2). Frequency and Duration: Participants reported the frequency (number of sessions per week) and the duration (minutes per week) of their physical activity.

(3). Intensity of Physical Activity: Participants were also asked to specify how many minutes per week they engaged in physical activity at an intensity level that induced heavy breathing and sweating.

The total physical amount of activity for each participant was calculated by summing up the durations of all reported activities. Based on their responses, participants were categorized into two levels of physical activity:

Low levels of Physical Activity: Defined as exercising 0–2 times per week.

High levels of Physical Activity: Defined as exercising more than 3 times per week.

This categorization allowed for the stratification of participants based on their level of engagement in physical activity, facilitating further analysis of its impact.

### 2.4. Long COVID Symptoms

Information on long COVID symptoms was gathered from participants who had contracted SARS-CoV-2 during the first two years of the COVID-19 pandemic. Participants were first asked if they had contracted COVID-19 (validated by self-antigen tests or health maintenance organization tests for positive COVID-19). Those who reported having been infected were subsequently asked whether they had experienced symptoms that persisted for more than three months after the acute phase of SARS-CoV-2 infection. In accordance with the World Health Organization (WHO) definition, a symptom was classified as long COVID only if the symptom was newly appearing, persisting over three months following acute infection of COVID-19, and lasting for at least 2 months [24]. Participants were asked to report which symptoms persisted for more than three months following their initial infection. The long COVID symptoms included the following: fatigue/exhaustion/concentration difficulties, memory impairment/confusion, feelings of sadness/low mood/depression, irritability/anxiety, altered/loss of sense of taste and smell, sleep disturbances, coughing, headache, sore throat, abdominal pain, gastrointestinal problems, diarrhea, muscle pain, decreased physical fitness, chest pain, palpitations, rapid heartbeat, shortness of breath at rest or during daily activities, and abnormal shortness of breath during exertion. Additionally, the total number of long COVID symptoms experienced by each participant was calculated.

### 2.5. Symptoms During the Acute COVID-19 Infection

Participants were asked to report the symptoms they experienced during their acute illness with SARS-CoV-2 (COVID-19). The listed symptoms included: fever above 37.5 °C lasting up to 2 days, fever above 37.5 °C lasting more than 3 days, fatigue or weakness, muscle ache, headache, reduced sense of taste and smell, shortness of breath, coughing, rhinorrhea, sore throat, gastrointestinal discomfort, abdominal pain, gastrointestinal problems, diarrhea, and any other symptoms. For each participant, the total number of reported acute COVID-19 symptoms was totaled.

Additional variables collected and used for adjustment of the statistical models include age, gender, number of times the participant was infected with COVID-19, and the date of the first COVID-19 contraction.

### 2.6. Statistical Analysis

Descriptive statistics were conducted to characterize the study population. Means, standard deviations, median, minimum, maximum, and frequency were calculated for the various variables. The study population was divided into two groups based on the amount of physical activity performed: low physical activity (defined as 0–2 times per week) and high physical activity (defined as more than three times per week).

Independent samples *t*-tests were used to compare mean physical activity levels and symptom counts between groups. Chi-square tests were applied to compare categorical variables. Pearson correlation coefficients were computed to examine associations between physical activity, acute infection symptoms, and long COVID symptoms. Normality of the continuous variables was assessed using skewness and kurtosis values, which fell within acceptable ranges (skewness < |1|, kurtosis < |1|), supporting the use of parametric tests. Statistical significance was set at *p* < 0.05. A multiple linear regression analysis was conducted to examine the extent to which demographic variables, physical activity, and acute COVID-19 symptom severity predicted the number of current long COVID symptoms reported. SPSS statistic version 29 software was used to perform the analyses.

## 3. Results

### Research Sample

During the course of the study, an online questionnaire was distributed to 2830 students enrolled in a college of education. The students who reported having contracted SARS-CoV-2 during the first two and a half years of the COVID-19 pandemic were prompted to complete the remainder of the questionnaire. A total of 309 students, with a median age of 26 years (ranging from 19 to 57 years), completed the questionnaire. Most respondents were female (83.2%). Among them, 72.5% had contracted one COVID-19 infection, and 69.6% were infected during the Omicron outbreak. Nearly 30% of the respondents were students of physical education, while the remainder were pursuing degrees in other disciplines (Figure 1, Table 1).

Among the respondents, 265 (85.8%) reported not experiencing long COVID symptoms, while 44 (14.4%) reported suffering from long COVID symptoms. Of those who are suffering from long COVID symptoms 265 (68.3%) are not students of physical education, and 84 (31.7%) are. Of the 44 reporting to be suffering from long COVID symptoms, 38 are not students of physical education and 6 are (Figure 1).

The most common long COVID symptom among the students was fatigue, reported by 13.3% of those surveyed. This was followed by elevated heart rate (12%), muscle pain, and shortness of breath—each reported by 9.4% of the participants. All other symptoms were reported by less than 7% of the population studied. Overall, the average number of long COVID symptoms among 54.7% of those who experienced long COVID was 1.5 types of symptoms (Figure 2).

There was no significant difference in the average number of long COVID symptoms between students who engage with low levels of physical activity and those who engage with high levels of physical activity (1.75 (0.89) vs. 1.83 (0.85) *p* = 0.376). Moreover, no difference was found in the weekly minutes of physical activity between students with long COVID and those without (*p* = 0.26). Additionally, there was no difference in the weekly time spent on physical activity that induce sweating and heavy breathing between the two groups (*p* = 0.272). When comparing physical education students to students in other disciplines, we found that physical education students experienced significantly fewer long COVID symptoms (6.7% vs. 17.4%) (Table 2).

A Pearson correlation analysis revealed a moderate correlation between the mean number of acute COVID-19 symptoms and long COVID symptoms (*p* = 0.492, *p* < 0.001). Additionally, weaker correlations were found in females (*p* = 0.144, *p* = 0.012) and in students who are not from the discipline of physical education (*p* = 0.129, *p* = 0.024) (Table 3).

A multiple linear regression analysis was conducted to examine the extent to which demographic variables, physical activity, and acute COVID-19 symptom severity predicted the number of current long COVID symptoms reported. The overall model was statistically significant, F (8, 97) = 5.060, *p* < 0.001, explaining 29.4% of the variance in symptom reporting (R^2^ = 0.294, adjusted R^2^ = 0.236). Among all predictors, only the mean number of acute COVID-19 symptoms during the infection emerged as statistically significant (B = 1.480, SE = 0.321, β = 0.422, *p* < 0.001), indicating that individuals who experienced more symptoms during their acute illness tended to report more long COVID symptoms. Other variables, including gender, physical activity level, academic program (e.g., physical education student), marital status, year of study, age, and number of children, were not significant predictors in the model.

## 4. Discussion

In this study, we aimed to characterize the prevalence of self-reported long COVID symptoms among undergraduate students of education and explore whether self-reported engagement in regular physical activity was associated with a reduced number of symptoms. In line with previous studies, we found that 14% of students reported that they suffer from long COVID. This finding is consistent with the literature, which suggests a prevalence rate of 7–30% among individuals who contracted COVID-19 without hospitalization [2,4]. Our research, like other studies, also revealed that females were more likely to experience long COVID symptoms [3,5,6,25,26,27]. Additionally, the more severe the acute phase of the COVID-19 infection, the more long COVID symptoms were reported by the students. This finding aligns with other research showing that the severity of acute illness is positively associated with the likelihood of developing long COVID [2,6]. The most prevalent symptom reported among the students was fatigue, which is also consistent with numerous studies that identify this as the most frequent symptom of long COVID [2].

In this study, no significant differences were found in the amount or intensity of physical activity between students who reported experiencing long COVID symptoms and those who did not. This finding contrasts with previous studies suggesting that regular physical activity may reduce the number or severity of long COVID symptoms [16,17,19]. However, many of those studies are observational in nature and cannot establish causality [28]. Furthermore, they often focus on recovery or symptom improvement, rather than on prevention of long COVID. Differences in study design, population characteristics, definitions of long COVID, and reliance on self-reported data may also contribute to inconsistent findings across studies. In our analysis, the only variable significantly associated with long COVID symptoms was the severity of the acute COVID-19 infection, supporting the notion that the intensity of the initial illness may be a stronger predictor of long COVID than physical activity levels [29]. Interestingly, although no significant difference in physical activity levels was observed, students enrolled in physical education programs reported fewer long COVID symptoms compared to peers in other academic tracks. These students are likely to engage in more frequent and intense physical activity as part of their academic routine, though further investigation is required to better understand this association.

## 5. Limitations

This study has several limitations. First, the sample size is relatively small, though statistically significant findings were obtained. Additionally, the research was conducted at a single college, which limits the generalizability of the findings. Replicating the study at other higher-education institutions would help validate the findings. Another limitation is the reliance on self-reported data, which may be subject to recall bias or inaccurate reporting, particularly regarding COVID-19 diagnosis, symptom severity, and physical activity levels. Furthermore, examining a larger cohort of students of physical education would allow for a more detailed analysis of whether variations in long COVID symptoms exist based on differing levels of physical activity within this population.

## 6. Conclusions

This study does not present an association between physical activity levels and a reduction in the number of long COVID symptoms. However, it did find that females reported a higher prevalence of symptoms than males, and that students enrolled in physical education programs—who are regularly engaged in structured physical activity—reported fewer long COVID symptoms than their peers in other academic disciplines. While these findings suggest a potential relationship worth exploring, the cross-sectional design limits conclusions about causality. Future longitudinal studies are needed to better understand these associations and to determine whether promoting physical activity through educational programs may support students’ recovery and long-term well-being.

## Figures and Tables

**Figure 1 ijerph-22-00754-f001:**
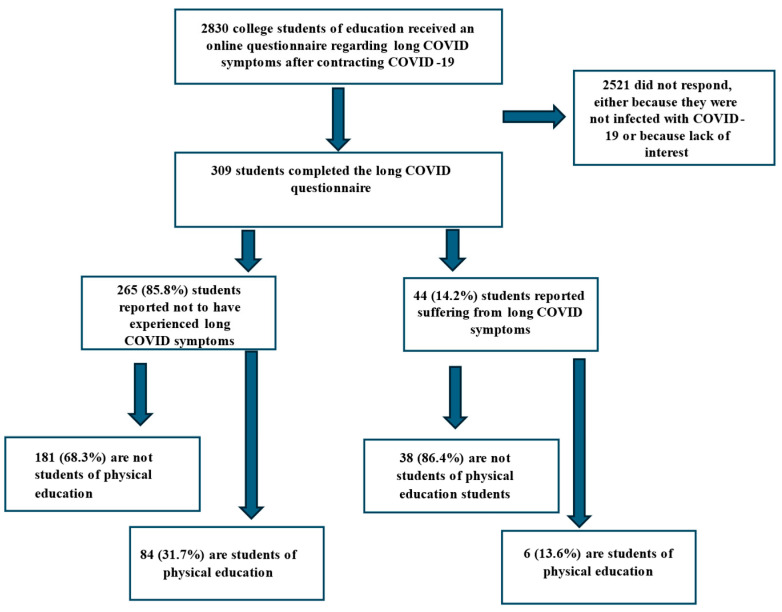
Research sample and incidence of long COVID.

**Figure 2 ijerph-22-00754-f002:**
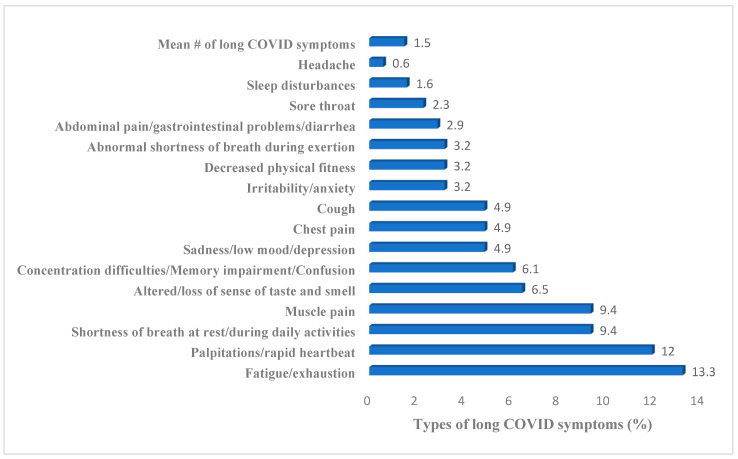
Different symptoms of long COVID in students suffering from long COVID (%).

**Table 1 ijerph-22-00754-t001:** Characteristics of the students who participated in the study.

Variables	
Age, (median, (min, max))	26, (19, 57)
Gender, female (N, %)	257, 83.2%
Status, single (N, %)	213. 68.9%
Study year (N, %)	
1st year	71, 23.0%
2nd year	63, 20.4%
3rd year	80, 25.9%
4th year	26, 8.4%
Other (M.Ed. studies/supplementing)	69, 22.3%
Students from disciplines other than physical education (N, %)	219, 70.9%
Contracted COVID-19 once (N, %)	224, 72.5%
Infected during the Omicron outbreak (N, %)	215, 69.6%
Number of symptoms during acute SARS-CoV-2 infection (mean, SD)	1.53, 0.22
Students reporting to be suffering from long COVID symptoms (N, %)	44/309, 14.24%

**Table 2 ijerph-22-00754-t002:** Differences in the number of long COVID symptoms depending on the amount, time, intensity of physical activity and the type of studies.

	N	Mean (SD)	Sig.
		Mean (SD) of long COVID symptoms	
Students who engage in a low level of physical activity	158	1.75 (0.89)	*p* = 0.376
Students who engage in a high level of physical activity	151	1.83 (0.85)	
	N	Mean (SD) Time of physical activity (3 = 61–120 min, 4 = 121–240 min)	Sig.
Weekley physical activity time (minutes) of students without symptoms of long COVID	263	3.75 (1.56)	*p* = 0.26
Weekley physical activity time (minutes) of students with symptoms of long COVID	44	3.86 (1.69)	
	N	Mean (SD) physical activity time (3 = 31–60 min, 4 = 61–120 min)	Sig.
Weekley amount of physical activity that causes sweating and strenuous breathing (minutes) of students without symptoms of long COVID	252	3.0 (1.5)	*p* = 0.272
Weekley amount of physical activity that causes sweating and strenuous breathing (minutes) of students with symptoms of long COVID	40	3.28 (1.22)	
	Students of Physical Education (N, %)	Education students from other disciplines (N, %)	
Students without symptoms of long COVID	89 (93.3%)	181 (82.6%)	
Students without symptoms of long COVID	6 (6.7%)	38 (17.4%)	

**Table 3 ijerph-22-00754-t003:** Pearson correlation for physical activity, average of COVID-19 symptoms, age, gender, marital status, type of studies and number of long COVID symptoms.

		Mean of Acute COVID-19 Infection Symptoms	Mean # of Long COVID Symptoms	Mean # of Times of Physical Activity Per Week	Age	Marital Status	Gender (Female)	Students Not Studying Physical Education Student
Mean # of acute COVID-19 infection symptoms	Pearson correlation Sig. (2-tailed)	1	0.492 *p* < 0.001	0.099 *p* = 0.082	0.69 *p* = 0.228	0.66 *p* = 0.263	0.144 *p* = 0.012	0.081 *p* = 0.155
Mean long COVID symptoms	Pearson correlation Sig. (2-tailed)		1	0.081 *p* = 0.154	0.018 *p* = 0.751	0.66 *p* = 0.920	0.118 *p* = 0.039	0.129 *p* = 0.024

## Data Availability

All data will be available upon request from the corresponding author.

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
