# Peer review of "The Impact of Physical Activity on Long COVID Symptoms Among College Students: A Retrospective Study"

_ijerph, 2025, doi:10.3390/ijerph22050754_

Round 1
Reviewer 1 Report
Comments and Suggestions for Authors
This is a well written work, but there are some key limitations that must be addressed.
Major Comments:
- The association between physical education students and decreased number of LC symptoms may be due to the fact that more men tend to be physical education teachers compared to teachers in other subjects. Male sex was also inversely related to the number of LC symptoms in your correlation in Table 3. To support the conclusion that physical education students report a fewer numbers of LC symptoms due to “greater awareness of a healthy lifestyle,” you must control for sex; otherwise, nothing definitive can be stated. A partial correlation or multinomial regression model would be appropriate.
- There is no discussion of the distribution of data, but parametric tests are used for comparison in Table 2. Either analyze and discuss the distribution of the data or utilize non-parametric tests of the median in Table 2.
- On line #97 in the methods, you report the number of students with LC to be 181, but then in the results line #185 the number reported is 44. Which is it?
Minor Comments:
- I am curious if mean or median physical activty differed by LC symptom reported in Figure 2. In other words, consider providing physical activity data (i.e., mean with 95% CI or Median with interquartile range) for each symptom listed in Figure 2.
- Change the phraseology in work from “decreased long – COVID symptoms” to “decreased number of long – COVID symptoms” where applicable. The former expression is unclear in the sense that decreased symptoms could suggest decreased intensity, duration, or number of symptoms.
Author Response
Dear Reviewer,
We thank you for your efficient and constructive review. Below are the answers relating to all the comments:
This is a well written work, but there are some key limitations that must be addressed.
Major Comments:
Comment 1: The association between physical education students and decreased number of LC symptoms may be due to the fact that more men tend to be physical education teachers compared to teachers in other subjects. Male sex was also inversely related to the number of LC symptoms in your correlation in Table 3. To support the conclusion that physical education students report a fewer numbers of LC symptoms due to “greater awareness of a healthy lifestyle,” you must control for sex; otherwise, nothing definitive can be stated. A partial correlation or multinomial regression model would be appropriate.
Response to Comment 1:
Thank you for this important observation. We fully agree that the potential confounding role of sex must be carefully considered when interpreting the association between being a physical education (PE) student and the number of reported long COVID (LC) symptoms. In response to this comment, we conducted multiple linear regression analyses for the full sample, as well as separately for PE students and non-PE students.
In the regression model conducted for the entire sample, the only statistically significant predictor of the number of reported LC symptoms was the mean number of acute COVID-19 symptoms experienced during the illness (B = 1.480, SE = 0.321, β = .422, t = 4.617, p < .001). This suggests that individuals who reported more symptoms during their acute infection tended to experience a greater number of ongoing symptoms. Notably, gender was not a statistically significant predictor in the model (B = 0.152, SE = 0.257, β = 0.056, p = .555), nor was the variable indicating whether the respondent was a PE student (B = 0.336, SE = 0.210, β = 0.152, p = .113).
The regression model for the full cohort was statistically significant, F(8, 97) = 5.060, p < .001, explaining approximately 29.4% of the variance in LC symptom reporting (R² = .294; adjusted R² = .236; standard error of the estimate = 0.872). Separate regression models conducted within the subgroups of PE students and non-PE students did not reveal any additional significant predictors.
These findings will be added to the Results section for clarity. While gender did show an inverse correlation with symptom count in the bivariate analysis (as noted in Table 3), it did not emerge as a significant predictor when controlling for other variables in the multivariate model. Therefore, although gender may play a role, it does not account for the association between PE student status and LC symptoms in this dataset. We acknowledge the importance of this issue and have revised the manuscript accordingly to clarify the interpretation and limitations of our findings.
Comment 2: There is no discussion of the distribution of data, but parametric tests are used for comparison in Table 2. Either analyze and discuss the distribution of the data or utilize non-parametric tests of the median in Table 2.
Response to Comment 2:
We thank the reviewer for this important observation. We examined the distribution of the relevant variables (weekly physical activity time and physical activity time that causes sweating and strenuous breathing) using descriptive statistics, including skewness and kurtosis values.
The skewness and kurtosis statistics for these variables were as follows:
- Weekly physical activity time (minutes PA now): Skewness = 0.285, Kurtosis = -0.949
- Weekly strenuous physical activity time (minutes PA sweat now): Skewness = -0.221, Kurtosis = -0.765
Both skewness values are well within the commonly accepted range of ±0.5, indicating approximately symmetric distribution. Kurtosis values are also within acceptable limits (±1), suggesting no severe departures from normality. Based on these results, the distributions can reasonably be considered approximately normal.
Consequently, we proceeded with parametric tests (independent-samples t-tests) for group comparisons, as the assumptions for normality and equal variances (confirmed via Levene’s test) were adequately met.
However, we recognize the importance of transparency regarding distributional assumptions and thus added a brief statement in the Methods section of the manuscript to clarify this.
This is the sentence that was added to the statistical analysis in the methods section:” Normality of the continuous variables was assessed using skewness and kurtosis values, which fell within acceptable ranges (skewness < |0.5|, kurtosis < |1|), supporting the use of parametric tests. Statistical significance was set at p < 0.05.”
Comment 3: On line #97 in the methods, you report the number of students with LC to be 181, but then in the results line #185 the number reported is 44. Which is it?
Response to Comment 3:
This was corrected. The number 181 in line #97 was a typo. The correct number is 44 students with LC. This was corrected.
Minor Comments:
Comment 4: I am curious if mean or median physical activity differed by LC symptom reported in Figure 2. In other words, consider providing physical activity data (i.e., mean with 95% CI or Median with interquartile range) for each symptom listed in Figure 2.
Response to Comment 4:
Thank you for your comment. We examined whether levels of physical activity differed based on the presence or absence of each of the Long-COVID symptoms presented in Figure 2. The analyses revealed no statistically significant differences in physical activity levels associated with the presence or absence of any individual LC symptom. As a result, we did not include these comparisons in the manuscript.
Comment 5: Change the phraseology in work from “decreased long – COVID symptoms” to “decreased number of long – COVID symptoms” where applicable. The former expression is unclear in the sense that decreased symptoms could suggest decreased intensity, duration, number of symptoms.
Response to Comment 5:
Thank you for the correction. The term: “decreased long – COVID symptoms”, was changed to “decreased number of long – COVID symptoms” where applicable.
Reviewer 2 Report
Comments and Suggestions for Authors
This is a cross-sectional analysis of the prevalence, symptoms, and influence of physical activity (PA) on long-covid among college aged students. While these findings are likely of interest to readers of this journal, there are major revisions necessary to increase the validity of the manuscript.
General
- The results and discussion highlight differences in long-covid prevalence and symptoms among physical education vs. nonphysical education students, although the need for this analysis it is not mentioned until late in the manuscript. There needs to be justification and inclusion of why this analysis was completed earlier in the manuscript including the introduction and methodology. Further, there needs to be a removal throughout the manuscript of the assumption that physical education students are more aware of healthy behaviors compared to other education students as this was not proven within your study, rather this is an assumption of the research team. If there is evidence of this from other studies, they need to be cited
- There is a major limitation in the cross-sectional design in which you do not know the extent to which PA influenced the development or management of the reported long-covid symptoms, rather we only know their association at one point in time. There needs to be adjustments to the tone of the discussion to better fit within the scope of the findings
- There appears to already be research in PA and long-covid per the introduction and discussion, so there needs to be a justification as to why your study is different/needed. Is your population understudied? If yes, this needs to be justified early in the manuscript and considered within the discussion. I see a sentence about this in the abstract, but not anywhere else in the manuscript
Abstract
- There needs to be mention that PA and long- covid variables were self-reported
Line 18: remove “who have greater awareness of a healthy lifestyle” as this is an assumption and not collected within the study
Lines 19: this study is cross-sectional so you cannot say that PA did not reduce long – covid symptoms, rather “Greater self-reported participation in PA was not associated with less reported long – covid symptoms among college-aged students"
Lines 21-22: I don’t think these findings support more PA integration into college curriculum. I would just present the results and leave any interpretation for later in the manuscript
Introduction:
- Page 1 line 38: correct to “Long – COVID”
- Page 1-2 lines 40-65: these two paragraphs can be merged and reduced to remove redundancy and remain focused on PA impact on long – COVID related symptoms, as this is the scope of the manuscript. Lines 41-47 can be deleted. Lines 47-51 can be incorporated into lines 52-65 as these are saying the same thing.
- Page 2 lines 82-86: this last sentence is redundant and not needed
- Page 2 final 3 paragraphs: these sentences and paragraphs jump back and forth between talking about how PA influences long covid development and management, which are 2 different topics. For ease of the reader, I suggest separating these points into different paragraphs (1) how PA may prevent development of Long – COVID, and (2) how PA may help management Long – COVID once acquired.
Materials & Methods
- Page 2 lines 89-90: change “cross-sectional retrospective study” to “cross-sectional study” and add the years of survey completion while removing “two years after the emergence of SARAS-CoV-2 and following 5 major COVID-19 waves”.
- Page 2 lines 91-95: change to “this study aimed to investigate the prevalence, symptom characteristics, and impact of PA on existing Long – COVID among college-aged students”
- Page 3 lines 96-98: these sentences are not needed and can be removed
- Page 3 lines 99-103: the inclusion and exclusion criterion can be in one paragraph
- Page 3 lines 104-114: this paragraph does not feel needed as these points are described in greater detail in the next few paragraphs
- Page 3 lines 118-119: here it needs to be directly stated that these are all self-reported variables as not to be confused with objective measures of PA and Long – Covid. Additionally, clarify distribution of the surveys either via in-person or digital delivery
- Page 4 lines 140-141: how do you know they had SARS covid during the pandemic? Self-reported?
- Page 4 lines 154-164: I am unsure what these additional variables add. Given the focus of the paper is long – covid, I suggest deleting these symptom-related variables for acute infection and their discussion as it does not add significance to the findings
Results
- Page 4 lines 177-178: how did you specifically target students who had an acute SARS infection? How did you know who they were?
- Results 7 lines 213-216: this needs to be moved to the data analysis section
- Page 7 lines 217-219: these sentences are not needed
- Page 7 line 221 and 222: use the Pearson correlation “p” rather than spell out the words
- Page 7 lines 224-226: this is an interpretation whereas the results section should just present objective findings
Discussion
- Page 8 line 245: this is an important section to re-iterate the self-reported nature of the variables
- Page 8 lines 244-270: this is a long paragraph with confusing point. This should be reduced to solely talking about the lack of association between amount of PA and long – covid symptoms. Be sure to remove any discussion of “physical fitness” as you did not complete a V02 max assessment
- Page 8 255-257: these study aspects don’t limit the studies generalizability, but rather are likely the contributing reason as to why your findings differ from other studies
Limitations
- Additional limitation is that you only conducted within one college – not just a single university
- Additional limitation is this is self-reported vs objective long-covid symptoms and PA which significantly decreases the validity of the findings
Conclusion
- Page 8 line 288: do not need to state “in conclusion”
- Page 8 line 288: change “correlation” to “association”
Author Response
Dear reviewer,
We thank you for your efficient and constructive review. Below are the answers relating to all the comments:
Comments and Suggestions for Authors
This is a cross-sectional analysis of the prevalence, symptoms, and influence of physical activity (PA) on long-covid among college aged students. While these findings are likely of interest to readers of this journal, there are major revisions necessary to increase the validity of the manuscript.
General
Comment 1. The results and discussion highlight differences in long-covid prevalence and symptoms among physical education vs. nonphysical education students, although the need for this analysis it is not mentioned until late in the manuscript. There needs to be justification and inclusion of why this analysis was completed earlier in the manuscript including the introduction and methodology. Further, there needs to be a removal throughout the manuscript of the assumption that physical education students are more aware of healthy behaviors compared to other education students as this was not proven within your study, rather this is an assumption of the research team. If there is evidence of this from other studies, they need to be cited.
Response to Comment 1:
Thank you for this important comment.
A justification was added at the end of the introduction: “Additionally, it investigates whether differences exist between students enrolled in general academic programs and those studying physical education, a discipline in which physical activity is an integral part of the curriculum, resulting in higher frequency and intensity of exercise among the latter group.
The assumption that physical education students are more aware of healthy behaviors compared to other education students was deleted throughout the manuscript (abstract, introduction and discussion).
Comment 2. There is a major limitation in the cross-sectional design in which you do not know the extent to which PA influenced the development or management of the reported long-covid symptoms, rather we only know their association at one point in time. There needs to be adjustments to the tone of the discussion to better fit within the scope of the findings.
Response to Comment 2:
Thank you for this insightful comment. The discussion section has been revised to appropriately reflect the cross-sectional nature of the study, with careful attention to framing the findings as associations rather than implying causality.
Comment 3. There appears to already be research in PA and long-covid per the introduction and discussion, so there needs to be a justification as to why your study is different/needed. Is your population understudied? If yes, this needs to be justified early in the manuscript and considered within the discussion. I see a sentence about this in the abstract, but not anywhere else in the manuscript.
Response to Comment 3:
Thank you for this important comment. A sentence was added at the end of the introduction, and discussion:
End of Introduction: “Notably, young adults have been found to report Long-COVID symptoms more frequently than older populations (2). However, the college student population—representing a significant subset of young adults—has been relatively underexplored in the context of Long-COVID, despite their vulnerability and the potential impact on academic functioning and quality of life. This study therefore aims to examine the prevalence and characteristics of Long-COVID among college students. Additionally, it investigates whether differences exist between students enrolled in general academic programs and those studying physical education, a discipline in which physical activity is an integral part of the curriculum, resulting in higher frequency and intensity of exercise among the latter group.”
Discussion: “Interestingly, although no significant difference in physical activity levels was observed, students enrolled in physical education programs reported fewer Long-COVID symptoms compared to peers in other academic tracks. These students are likely to engage in more frequent and intense physical activity as part of their academic routine, though further investigation is required to better understand this association.”
Abstract
Comment 4. There needs to be mention that PA and long- covid variables were self-reported
Response to Comment 4:
Thank you for the comment. This is the new sentence inserted in the abstract: “ Greater self-reported participation in physical activity was not associated with less reported Long-COVID symptoms among college-aged students; however, students enrolled in physical education programs—who integrate physical activity into their daily routines as part of their academic curriculum—reported fewer symptoms, suggesting that sustained, structured physical activity may contribute to reduced symptom burden.
Comment 5: Line 18: remove “who have greater awareness of a healthy lifestyle” as this is an assumption and not collected within the study
Response to Comment 5:
Thank you for the important comment. The sentence was removed. This is the new sentence integrated in the abstract: “Greater self-reported participation in physical activity was not associated with less reported Long-COVID symptoms among college-aged students; however, students enrolled in physical education programs—who integrate physical activity into their daily routines as part of their academic curriculum—reported fewer symptoms, suggesting that sustained, structured physical activity may contribute to reduced symptom burden.”
Comment 6: Lines 19: this study is cross-sectional so you cannot say that PA did not reduce long – covid symptoms, rather “Greater self-reported participation in PA was not associated with less reported long – covid symptoms among college-aged students"
Response to Comment 6:
The sentence was changed as recommended: “Greater self-reported participation in physical activity was not associated with less reported Long-COVID symptoms among college-aged students; however, students enrolled in physical education programs—who integrate physical activity into their daily routines as part of their academic curriculum—reported fewer symptoms, suggesting that sustained, structured physical activity may contribute to reduced symptom burden. Further research is needed to explore this relationship.”
Comment 7: Lines 21-22: I don’t think these findings support more PA integration into college curriculum. I would just present the results and leave any interpretation for later in the manuscript.
Response to Comment 7:
The sentence was deleted, and changed to: “Greater self-reported participation in physical activity was not associated with less reported Long-COVID symptoms among college-aged students; however, students enrolled in physical education programs—who integrate physical activity into their daily routines as part of their academic curriculum—reported fewer symptoms, suggesting that sustained, structured physical activity may contribute to reduced symptom burden.”
- Introduction:
Comment 8: - Page 1 line 38: correct to “Long – COVID”
Response to Comment 8:
This was Corrected.
Comment 9: - Page 1-2 lines 40-65: these two paragraphs can be merged and reduced to remove redundancy and remain focused on PA impact on long – COVID related symptoms, as this is the scope of the manuscript. Lines 41-47 can be deleted. Lines 47-51 can be incorporated into lines 52-65 as these are saying the same thing.
Response to Comment 9:
Thank you for your helpful comment. The two paragraphs were merged into one:
“Physical activity improves multiple physiological systems, it improves metabolic functions, enhances endurance, and supports healthy weight maintenance [7,8]. It also reduces the risk of depression and anxiety [9,10], and may strengthen immune responses, potentially protecting against severe infections and prolonged recovery such as in Long-COVID [11–13]. The various health benefits of regular physical activity suggest that might help manage Long - COVID symptoms like autonomic dysfunction and fatigue [13,14], as well as reduce anxiety and depression while improving cognitive function, potentially supporting Long - COVID patients experiencing neurocognitive symptoms [15,16]. Moreover, studies suggest that physical activity during the pandemic reduced the likelihood and duration of Long - COVID symptoms [17,18], and pre-infection physical activity was linked to a lower risk of post-COVID self-care difficulties [19]. In addition, individuals engaging in higher physical activity levels before COVID-19 had a lower risk of severe COVID-19 outcomes [20].”
Comment 10: - Page 2 lines 82-86: this last sentence is redundant and not needed.
Response to Comment 10:
The sentences were removed.
Comment 11: - Page 2 final 3 paragraphs: these sentences and paragraphs jump back and forth between talking about how PA influences long covid development and management, which are 2 different topics. For ease of the reader, I suggest separating these points into different paragraphs (1) how PA may prevent development of Long – COVID, and (2) how PA may help management Long – COVID once acquired.
Response to Comment 11:
Thank you for this helpful comment. The paragraph was rewritten:
“While regular physical activity may offer benefits for both acute and long-term COVID-19 symptoms in non-hospitalized individuals, current findings remain inconclusive due to the complex nature of Long-COVID and heterogeneity in study methodologies (16 - 22). Notably, young adults have been found to report Long-COVID symptoms more frequently than older populations (2). However, the college student population—representing a significant subset of young adults—has been relatively underexplored in the context of Long-COVID, despite their vulnerability and the potential impact on academic functioning and quality of life. This study therefore aims to examine the prevalence and characteristics of Long-COVID among college students. Additionally, it investigates whether differences exist between students enrolled in general academic programs and those studying physical education, a discipline in which physical activity is an integral part of the curriculum, resulting in higher frequency and intensity of exercise among the latter group.”
- Materials & Methods
Comment 12: - Page 2 lines 89-90: change “cross-sectional retrospective study” to “cross-sectional study” and add the years of survey completion while removing “two years after the emergence of SARAS-CoV-2 and following 5 major COVID-19 waves”.
Response to Comment 12:
Thank you for this comment. The sentence was changed to:
"This cross-sectional study was conducted between December 2022 and March 2023. During this period, a survey was distributed via Google Forms to 2,830 students enrolled in a Bachelor of Education (B.Ed.) program at a college of education."
Comment 13: - Page 2 lines 91-95: change to “this study aimed to investigate the prevalence, symptom characteristics, and impact of PA on existing Long – COVID among college-aged students”
Response to Comment 13:
The sentence was changed to:” this study aimed to investigate the prevalence, symptom characteristics, and impact of PA on existing Long – COVID among college-aged students”, as recommended.
Comment 14: - Page 3 lines 96-98: these sentences are not needed and can be removed.
Response to Comment 14:
The sentences were removed.
Comment 15: - Page 3 lines 99-103: the inclusion and exclusion criterion can be in one paragraph.
Response to Comment 15:
The inclusion and exclusion were integrated to one paragraph.
Comment 16: - Page 3 lines 104-114: this paragraph does not feel needed as these points are described in greater detail in the next few paragraphs
Response to Comment 16:
Lines 104-114 were removed
Comment 17: - Page 3 lines 118-119: here it needs to be directly stated that these are all self-reported variables as not to be confused with objective measures of PA and Long – Covid. Additionally, clarify distribution of the surveys either via in-person or digital delivery
Response to Comment 17:
This sentence was added: All questionnaires were self-reported and distributed via digital delivery.
Comment 18: - Page 4 lines 140-141: how do you know they had SARS covid during the pandemic? Self-reported?
Response to Comment 18:
Yes, COVID-19 infection was self-reported; however, participants were instructed to indicate a positive infection only if it had been confirmed by a self-administered antigen test or an official test conducted through a health maintenance organization (HMO). At the time, public health policies strongly encouraged testing, resulting in high compliance across the population.
The sentence was changed to: “Participants were first asked if they had contracted COVID-19 (validated by self antigen tests or health maintenance organization tests for positive COVID-19). Those who reported having been infected were subsequently asked whether they had experienced symptoms that persisted for more than three months after the acute phase of SARS-CoV-2 infection.”
Comment 19: - Page 4 lines 154-164: I am unsure what these additional variables add. Given the focus of the paper is long – covid, I suggest deleting these symptom-related variables for acute infection and their discussion as it does not add significance to the findings
Response to Comment 19:
The severity of symptoms during the acute phase of COVID-19 was found to be associated with the severity of Long COVID symptoms and was therefore included as a variable in the analysis. This important information was added to the introduction: “Interestingly, adults under 50 report a higher prevalence of Long COVID symptoms compared to those over 50, and evidence suggests that greater severity of symptoms during the acute phase of COVID-19 is associated with an increased likelihood of developing Long COVID [2].
- Results
Comment 20: - Page 4 lines 177-178: how did you specifically target students who had an acute SARS infection? How did you know who they were?
Response to Comment 20:
Thanks for the comment. The sentence was changed to make it clearer: “The students who reported having contracted SARS-CoV-2 during the first two and a half years of the COVID-19 pandemic; were prompted to complete the remainder of the questionnaire."
Comment 21: - Results 7 lines 213-216: this needs to be moved to the data analysis section
Response to Comment 21:
The sentences were removed.
Comment 22: - Page 7 lines 217-219: these sentences are not needed.
Response to Comment 22:
The sentences were removed.
Comment 23: - Page 7 line 221 and 222: use the Pearson correlation “p” rather than spell out the words.
Response to Comment 23:
The words Pearson correlation were changed to “p”.
Comment 24: - Page 7 lines 224-226: this is an interpretation whereas the results section should just present objective finding.
Response to Comment 24:
The Sentences were removed.
- Discussion
Comment 25: - Page 8 line 245: this is an important section to re-iterate the self-reported nature of the variables
Response to Comment 25:
Thank you for this helpful comment. In response, we have revised the sentence to highlight the self-reported nature of the variables, as follows:
“In this study, we aimed to characterize the prevalence of self-reported Long-COVID symptoms among undergraduate students of education and explore whether self-reported engagement in regular physical activity was associated with a reduced number of symptoms.”
Comment 26: - Page 8 lines 244-270: this is a long paragraph with confusing point. This should be reduced to solely talking about the lack of association between amount of PA and long – covid symptoms. Be sure to remove any discussion of “physical fitness” as you did not complete a V02 max assessment
Response to Comment 26:
Thank you for this helpful comment. The paragraph was revised to address the concerns raised. It now reads: “In this study, no significant differences were found in the amount or intensity of physical activity between students who reported experiencing Long-COVID symptoms and those who did not. This finding contrasts with previous studies suggesting that regular physical activity may reduce the number or severity of Long-COVID symptoms [17,18,20]. However, many of those studies are observational in nature and cannot establish causality [29]. Furthermore, they often focus on recovery or symptom improvement, rather than on prevention of Long-COVID. Differences in study design, population characteristics, definitions of Long-COVID, and reliance on self-reported data may also contribute to inconsistent findings across studies. In our analysis, the only variable significantly associated with Long-COVID symptoms was the severity of the acute COVID-19 infection, supporting the notion that the intensity of the initial illness may be a stronger predictor of Long-COVID than physical activity levels [30].”
Comment 27: - Page 8 255-257: these study aspects don’t limit the studies generalizability, but rather are likely the contributing reason as to why your findings differ from other studies
Response to Comment 27:
Thank you for this important comment. The sentence was changed as follows: Additionally, variations in how Long - COVID is defined, differences in populations studied, and reliance on self-reported data might differ our findings from other studies.
- Limitations
Comment 28: - Additional limitation is that you only conducted within one college – not just a single university
Response to Comment 28:
Thank you for the comment. This was added: “Additionally, the research was conducted at a single College, which limits the generalizability of the findings. Replicating the study at other higher-education institutions would help validate the findings.”
Comment 29: - Additional limitation is this is self-reported vs objective long-covid symptoms and PA which significantly decreases the validity of the findings
Response to Comment 29:
Thank you for the comment. This limitation was added: “Another limitation is the reliance on self-reported data, which may be subject to recall bias or inaccurate reporting, particularly regarding COVID-19 diagnosis, symptom severity, and physical activity levels.”
- Conclusion
Comment 30: - Page 8 line 288: do not need to state “in conclusion”
Response to Comment 30:
The word conclusion was removed.
Comment 31: - Page 8 line 288: change “correlation” to “association”
Response to Comment 31:
The word correlation was changed to association.

Round 2
Reviewer 1 Report
Comments and Suggestions for Authors
All major elements have been addressed in my mind. But the references list contains some references that incorrectly are only hyperlinks - specifically #4, #8, #23, and #25.
These need to be corrected.
Author Response
Reviewer 1
We thank you for your prompt and constructive review. Please find our response to the comment regarding the references below:
All major elements have been addressed in my mind. But the references list contains some references that incorrectly are only hyperlinks - specifically #4, #8, #23, and #25.
These need to be corrected.
Response to comment:
The references noted (#4, #8, #23, and #25) have been revised and now appear in the correct format appropriate for a reference list:
4. World Health Organization. Post COVID-19 condition [Internet]. Copenhagen: WHO Regional Office for Europe; 2024 Feb 6 [cited 2025 May 7]. Available from: https://www.who.int/europe/news-room/fact-sheets/item/post-covid-19-condition
8. World Health Organization. Physical activity [Internet]. Geneva: World Health Organization; 2024 [cited 2025 May 7]. Available from: https://www.who.int/news-room/fact-sheets/detail/physical-activity
23. Joseph G, Schori H. The beneficial effect of the first COVID-19 lockdown on undergraduate students of education: Prospective cohort study. JMIR Form Res. 2022;6(2):e27286. doi:10.2196/27286
25. World Health Organization. Post COVID-19 condition [Internet]. Copenhagen: WHO Regional Office for Europe; 2022 Dec 7 [cited 2025 May 8]. Available from: https://www.who.int/europe/news-room/fact-sheets/item/post-covid-19-condition
Reviewer 2 Report
Comments and Suggestions for Authors
The research team appropriately addressed comments from revision 1. Remaining edits are very minor grammatical changes in the new text
Page 1 line 43: change comma to period, then capitalize “It” for starting new sentences
Page 4 lines 156-157: Remove colon after COVID-19 pandemic. Replace end of sentence comma with period
Page 7 line 234: add period after “than physical activity levels” to denote end of sentence
Author Response
Reviewer 2
We thank you for your prompt and constructive review. Please find our responses to your comments below:
Comment 1: Page 1, line 43
Change comma to period, then capitalize “It” for starting a new sentence.
Response to Comment 1:
The comma was changed to a period, and the word "It" was capitalized at the beginning of the sentence:
“Physical activity improves multiple physiological systems. It improves metabolic functions, enhances endurance, and supports healthy weight maintenance [7,8].”
Comment 2: Page 4, lines 156–157
Remove colon after “COVID-19 pandemic.” Replace end-of-sentence comma with a period.
Response to Comment 2:
The colon was removed, and the comma at the end of the sentence was replaced with a period:
“The students who reported having contracted SARS-CoV-2 during the first two and a half years of the COVID-19 pandemic were prompted to complete the remainder of the questionnaire.”
Comment 3: Page 7, line 234
Add period after “than physical activity levels” to denote end of sentence.
Response to Comment 3:
The sentence was corrected accordingly:
“In our analysis, the only variable significantly associated with Long-COVID symptoms was the severity of the acute COVID-19 infection, supporting the notion that the intensity of the initial illness may be a stronger predictor of Long-COVID than physical activity levels [30].”